# *DSP*-Related Cardiomyopathy as a Distinct Clinical Entity? Emerging Evidence from an Italian Cohort

**DOI:** 10.3390/ijms24032490

**Published:** 2023-01-27

**Authors:** Francesca Di Lorenzo, Enrica Marchionni, Valentina Ferradini, Andrea Latini, Laura Pezzoli, Annamaria Martino, Fabiana Romeo, Annamaria Iorio, Stefano Bianchi, Maria Iascone, Leonardo Calò, Giuseppe Novelli, Ruggiero Mango, Federica Sangiuolo

**Affiliations:** 1Department of Biomedicine and Prevention, University of Rome Tor Vergata, Via Montpellier 1, 00133 Rome, Italy; 2Medical Genetics, Policlinico Tor Vergata, 00133 Rome, Italy; 3Laboratory of Medical Genetics, ASST Papa Giovanni XXIII, 24127 Bergamo, Italy; 4Department of Cardiology, Policlinico Casilino, 00169 Rome, Italy; 5Cardiology Department, ASST Papa Giovanni XXIII Bergamo, 24127 Bergamo, Italy; 6UOC Cardiologia, Ospedale Fatebenefratelli Isola Tiberina, 00186 Rome, Italy; 7House of Care D4, Local Health Authority Roma 2, 00185 Rome, Italy

**Keywords:** desmoplakin, arrhythmogenic cardiomyopathy, dilated cardiomyopathy

## Abstract

Variants in desmoplakin gene (*DSP* MIM *125647) have been usually associated with Arrhythmogenic Cardiomyopathy (ACM), or Dilated Cardiomyopathy (DCM) inherited in an autosomal dominant manner. A cohort of 18 probands, characterized as heterozygotes for *DSP* variants by a target Next Generation Sequencing (NGS) cardiomyopathy panel, was analyzed. Cardiological, genetic data, and imaging features were retrospectively collected. A total of 16 *DSP* heterozygous pathogenic or likely pathogenic variants were identified, 75% (n = 12) truncating variants, n = 2 missense variants, n = 1 splicing variant, and n = 1 duplication variant. The mean age at diagnosis was 40.61 years (IQR 31–47.25), 61% of patients being asymptomatic (n = 11, New York Heart Association (NYHA) class I) and 39% mildly symptomatic (n = 7, NYHA class II). Notably, 39% of patients (n = 7) presented with a clinical history of presumed myocarditis episodes, characterized by chest pain, myocardial enzyme release, 12-lead electrocardiogram abnormalities with normal coronary arteries, which were recurrent in 57% of cases (n = 4). About half of the patients (55%, n = 10) presented with a varied degree of left ventricular enlargement (LVE), four showing biventricular involvement. Eleven patients (61%) underwent implantable cardioverter defibrillator (ICD) implantation, with a mean age of 46.81 years (IQR 36.00–64.00). Cardiac magnetic resonance imaging (CMRI) identified in all 18 patients a delayed enhancement (DE) area consistent with left ventricular (LV) myocardial fibrosis, with a larger localization and extent in patients presenting with recurrent episodes of myocardial injury. These clinical and genetic data confirm that *DSP*-related cardiomyopathy may represent a distinct clinical entity characterized by a high arrhythmic burden, variable degrees of LVE, Late Gadolinium Enhancement (LGE) with subepicardial distribution and episodes of myocarditis-like picture.

## 1. Introduction

Arrhythmogenic Cardiomyopathy (ACM) is an inherited disorder characterized by the progressive replacement of the ventricular myocardium by fibrofatty tissue, which predispose to develop ventricular arrhythmias. In contrast to what was previously thought, the disorder does not involve only the right ventricle (RV), but it may also affect the left ventricle (LV), or both, as reported in several studies [1,2,3,4,5,6]. Recently, an international expert consensus document proposed new criteria for the diagnosis of ACM (the “Padua criteria”), to cover all the phenotypic expressions of the disease [7]. Affected patients usually present arrhythmic symptoms, such as palpitations, chest tightness, and syncopes, and may present variable signs of heart failure (HF). Notably, the ACM phenotype can also overlap with other cardiomyopathies (CMPs), particularly dilated cardiomyopathy (DCM), in which arrhythmia may be present along with other distinctive phenotypic features such as ventricular dilation and depressed myocardial performance. In fact, according to the European Society of Cardiology (ESC), DCM is currently defined by the presence of left ventricular or biventricular dilatation and systolic dysfunction in the absence of abnormal loading conditions (hypertension, valve disease) or coronary artery disease, sufficient to cause global systolic impairment [8]. It is important to underline that both conditions are characterized by a remarkable phenotypic variability, which make complex the classification in clinically distinct entities [9,10]. Furthermore, biological studies of pathogenic mechanisms underlying the development of inherited cardiomyopathies have led to an awareness of final common protein pathways [11]. Although DCM is predominantly caused by variants in Titin (*TTN*) or Lamin A/C (*LMNA*) genes, genes encoding for desmosomal proteins are known to be involved, too [10,12]. Mutations in desmosomal genes are indeed the most common genetic cause of ACM [13,14,15].

Desmosomes are intercellular junctions, which provide strong adhesion between cells and link intracellularly to the intermediate filament of cytoskeleton, giving mechanical strength to tissues. They are mostly represented in tissues subjected to continuous mechanical stress, such as epidermidis and myocardium [16,17]. Desmoplakin, encoded by the *DSP* gene (MIM *125647), is the most prevalent component of desmosome. There are three different isoforms, produced by alternative splicing of the *DSP* gene, which differ from each other by the length of rod domain: a long isoform (DSP-I), with a whole rod domain; an intermediate isoform (DSP-Ia), including a half of rod domain; and a short isoform (DSP-II), which misses the majority of rod domain. DSP-I is the most prevalent cardiac isoform, which connects the cardiac desmosome to intermediate filaments, and it is crucial for force transmission in the myocardium [18].

Thus, variants in the *DSP* gene have been associated with both DCM and ACM phenotypes, especially left-dominant ventricular cardiomyopathy (ALVC), inherited in autosomal manner [19,20]. Variants causing a desmoplakin loss of function lead to the failure of intercellular adhesion and have been associated with histological findings of inflammatory infiltrates and fibrosis of the left ventricular myocardium [21,22]. Moreover, small *DSP* case series reported in these patients the occurrence of recurrent myocarditis, that may be the initial presentation of cardiomyopathy, preceding systolic dysfunction [23,24,25]. Considering these findings, which appear to differentiate *DSP*-related cardiomyopathy from classic right ventricular arrhythmogenic cardiomyopathy (ARVC) and DCM, some authors have suggested the possibility of a new distinct clinical entity characterized by a prevalent fibrotic and inflammatory component requiring its own specific clinic management [22,26]. 

With these assumptions in the present study, we retrospectively evaluated the clinical and instrumental characteristics of a selected cohort of patients carrying heterozygous *DSP* pathogenic (P), or likely pathogenic (LP) variants, to identify distinctive phenotypic features and rule out a possible genotype-phenotype correlation.

## 2. Results

### 2.1. Genetics Variants

A total of 16 heterozygous variants in the *DSP* gene were identified in 18 probands (Table 1) since the same variants were detected in two couples of two unrelated patients (rs869025395 in patients ID 5 and 17; rs1581819043 in patients ID 7 and 9). Of them, 75% (n = 12) were truncating variants, two were missense, one a splicing variant, and one a duplication variant. The majority of variants were located in “coiled coil domain” (n = 8), followed by “rod domain” (n = 4), and “spectrin domain” (n = 3) and “pectin repeat domain” (n = 1). More than half of the identified variants are located in the head of the protein. Of 16 variants, 10 were not reported in scientific literature nor in ClinVar [27]; five variants were cited in ClinVar [27]; seven were reported in the Single Nucleotide Polymorphism Database (dbSNP) [28]; seven were classified as “pathogenic” (P); and nine as “likely pathogenic” (LP) according to American College of Medical Genetics (ACMG) criteria [29]. Two patients (ID 9 and 16) also carried a heterozygous variant in the *DSC2* gene. Specifically, patient ID 9 carried the variant c.2381C>T, p.(Ser794Leu) and patient ID 16 carried the variant c.304G>A, p.(Glu102Lys). Both of these examples are cited in ClinVar as a variant of uncertain significance (VoUS)/likely benign variant and reported in Genomic Aggregation Database (gnomAD), with an allele frequency of 0.004 and 0.000777, respectively [27,30]. 

### 2.2. Demographics and Clinical Presentation of Probands

The demographics and genetics data of 18 unrelated individuals who carried LP or P variants in the *DSP* gene were collected (Table 2). The mean age at diagnosis was 40.61 years (IQR 31–47.25); the youngest patient was a girl who presented recurrent myocarditis-like episodes, as did the other three patients, by the age of 18 years. At the time of first genetic counseling, none of the probands have presented a major arrhythmic event (MAE), defined as fatal or life-threatening arrhythmic events, except two female patients (ID 1 and 16) who presented a sustained ventricular tachycardia (SVT) pharmacologically cardioverted. The majority of patients were asymptomatic (61%, n = 11, NYHA I) or mildly symptomatic (39%, n = 7, NYHA II). Palpitations were the most common reported symptoms (78%, n = 14), followed by chest pain (44%, n = 8), with 39% of patients (n = 7) showing both symptoms. Seven patients (39%) suffered from myocarditis-like episodes, in four cases (57%) they were recurrent. Finally, two patients (ID 5 and 13) presented a syncopal episode. Nine patients (50%) presented a family history of sudden cardiac death (SCD), eight patients (44%) presented a family history of DCM and/or ACM and four patients (22%) presented family history for both conditions. Notably, more than half of the patients (61%, n = 11) underwent ICD implantation.

### 2.3. Cardiac Imaging

All probands underwent a full cardiologic evaluation before they were referred to genetic counseling. Data from echocardiography and/or cardiac magnetic resonance imaging (CMRI) at first clinical assessment were retrospectively collected (Table 3). The majority of patients (61%, n = 11) presented with a mild decreased or at lower limits systolic function, frequently associated with cardiac wall abnormalities involving predominantly the left ventricle. At the time of diagnosis, slightly more than half of the patients (55%, n = 10) showed left ventricular enlargement (LVE) with a variable degree of ventricular dilation and four of them (40%) showed biventricular enlargement. Of note in all patients, CMRI lead to the identification of an area of delayed enhancement (DE) consistent with LV myocardial fibrosis, even if ventricular ejection fraction (VEF) was preserved. The most frequent localization of DE was subepicardial with left ventricular walls (LVW) and intraventricular septum involvement with a non-ischemic pattern. Particularly, in patient ID 1, CMRI showed morpho-functional modifications consistent with ACM with biventricular phenotype, while in probands ID 3 and 4, CMRI identified subepicardial DE distributed along postero-lateral and infero-lateral LVW, consistent with ALVC. In patients ID 7, 8, 9, and 10 affected by recurrent episodes of myocardial injury, CMRI showed a more extensive DE distribution with subepicardial or mid-subepicardial localization, as expected in a post-inflammatory cardiomyopathy (Appendix A). In patient ID 8 myocardial fibrosis was also confirmed by the histological exam of endomyocardial bioptic tissue, which showed several areas of polychronic fibrosis, surrounded by myocytes with clear and significant involutional aspects with the rarefaction and disaggregation of the contractile material, nuclear clearing or complete nuclear involution.

### 2.4. Electrocardiography, Arrythmias and ICD

At electrocardiography (ECG) baseline examination all probands showed an atrioventricular and intraventricular conduction substantially within normal limits (Table 4). Most individuals (67%, n = 12) presented QRS abnormalities, such as low voltage in peripheral leads (33%, n = 6), the fragmentation of QRS complex (fQRS) (28%, n = 5) and poor R wave progression (PRWP) (n = 1). Non-specific abnormalities (NSA) in ventricular repolarization (VR) were the most common anomalies reported (61%, n = 11). In two patients, ECG showed T wave inversion (TWI), respectively, in anterior and infero-lateral leads in patient ID 1 and inferior leads in patient ID 4. Holter ECG monitoring of 24 h is available in all patients, showing a high burden of ventricular ectopy (VE) with frequent or very frequent premature ventricular contractions (PVCs) of at least two distinct morphologies in 72% of individuals (n = 13). Of them, in eight cases (62%) they were organized in repetitive fashion, both couples and salvos. In two patients Holter ECG monitoring of 24 h revealed a low burden of VE with occasional polymorphic PVCs. Specifically, Holter ECG performed in patient ID 10, during treatment with metoprolol, showed some PVCs evenly distributed throughout the 24 h, mostly singles, some organized in couples, and two polymorphic triplets. An ECG stress test, later performed, showed some PVCs of different morphology at the peak of exercise and some monomorphic in the recovery phase. In patient ID 17, an ECG Holter showed some PVCs, mostly singles and monomorphic, some of different morphology and organized in couples. Of note, in patient ID 16, Holter ECG monitoring showed frequent polymorphic VE with some episodes of SVT, prior to define clinical and genetic diagnosis. ECG stress test data were available for four of these patients; none of them resulted positive for inducible ischemia. Particularly, an ECG stress test of patient ID 4 highlighted several PVCs of infundibular origin, both singles and organized in couples, predominantly in recovery phase; patient ID 5’s test revealed numerous PVCs, during exercise and in recovery phase, with left bundle branch and right bundle branch block morphology, some organized in couples, absents at peak exercise. In patient ID 8, an ECG stress test performed at first clinical assessment showed several, singles, polymorphic PVCs and three couples at the beginning of the exercise, which disappeared right after. Finally, in patient ID 17, and ECG stress test identified a symptomatic triplet at the peak of the exercise. A further test performed after 3 years in patient ID 8 showed in rest phase frequent, singles, polymorphic PVCs, with two couples and two triplets during exercise, persistent in recovery phase. 

As previously reported in Table 2, 11 patients (61%) underwent ICD implantation; in nine of them for primary prevention, and in ID 1 and ID 16 for secondary prevention after SVT. Mean age at ICD implantation was 46.81 years (IQR 36.00–64.00), after an average interval from diagnosis of 3.1 years (IQR 0.00–3.00). Nine patients underwent implantation of an intracardiac device, six bicameral (Dual-Chamber; D-C) and three monocameral (Single-Chamber; S-C), respectively, while the other two were implanted with subcutaneous ICD (S-ICD). At the time of ICD implantation, Left Ventricular Ejection Fraction (LVEF) mean value was 43.00% (IQR 35.00–50.00) (Table 5).

### 2.5. Segregation Data Analysis

After the identification of LP or P variants, genetic screening was offered to first degree relatives of affected probands. Data from the segregation analysis of three families (ID 2, ID 6, and ID 7) are available. 

In the family of patient ID 2, genetic testing was performed in both his two sons and in the only nephew, whose father, the probands’ brother, suddenly died at the age of 51 years (Figure 1). Segregation analysis was not performed on the probands’ parents and brother because they had already died (Figure 1). His first child (IV:1) resulted as a carrier for heterozygous pathogenic variant Q833X in the *DSP* gene. At the time of the genetic counseling, he was 22 years old and apparently asymptomatic, with no history of syncope, chest pain and/or palpitations. Of note, a recent cardiologic evaluation was not available for the patient, thus it is not possible to exclude an unknown underlying condition. The second child (IV: 2) and the nephew (IV: 3) resulted negative at genetic testing. 

In the family of patient ID 6, genetic testing was performed on her two daughters, which resulted in both being carriers of E1068Vfs*1 in the *DSP* gene (Figure 2). Her first daughter (IV:1), at 38 years old, received a full cardiologic evaluation, which resulted within the limits. Her second daughter (IV:2) was 28 years old at the time of genetic counselling and she did not undergo cardiological examination in the previous two years; neither of them reported any symptoms.

In the family of patient ID 7, genetic testing was performed on both probands’ parents and led to the identification of maternal segregation of P variant G1737Dfs*16 (Figure 3). The probands’ mother has a personal history of frequent ventricular extrasystoles in the absence of any signs of morpho-functional abnormalities at echocardiography performed during the previous year. 

## 3. Discussion

*DSP* variants have been recently reported to be causative of a form of cardiomyopathy characterized by recurrent myocardial injury episodes and left ventricular fibrosis [22]. In this cohort, seven patients reported a previous history of myocardial injury episodes and all of them presented DE at CMRI (Table 2 and Table 3). Extensive infectious and immunological screenings were performed in patients who suffered from presumed myocarditis episodes and both resulted negative. In one case (patient ID 8), a cardiac biopsy was performed, showing histological findings consistent with inflammatory cardiomyopathy, and resulting negative for viral markers. In this patient, myocardial injury was the first clinical presentation of disease along with frequent polymorphic PVCs. A short time after the first episodes, the patient presented a drastic decrease in systolic function, previously reported as normal or at the lower limits (Table 3), with a LVEF reduced by up to 25% and only partially recovered after appropriate clinical management, leading to S-ICD implantation at the age of 34 years (Table 5). Notably, even if they were not detected at the first clinical cardiological assessment (Table 4), performed when he was 31 years old, TWI in V5-V6 and low voltage QRS complex, the latter finding already known, were observed at the last ECG baseline, carried out at the age of 36 years (Appendix A). After 5 months from the first genetic counseling, the patient presented two episodes of SVT: one was treated with direct current (DC) shock, and the other resolved spontaneously. At CMRI, circumferential DE was observed. Similarly, patient ID 9 was referred for medical attention at the age of 18 years due to recurrent episodes of myocardial injury associated with chest pain and palpitations. Holter ECG registration detected frequent PVCs with two morphologies isolated and organized in 10 couples and one triplet (Appendix A). Differently from patient ID 8, LVEF was preserved at the time of genetic counselling, when she was 23 years old. CMRI detected an extensive and persistent mid- and subepicardial DE, involving 30% of myocardial mass (Appendix A). In contrast, in patient ID 7, no history of arrhythmias or others electrocardiographic anomalies was reported, nor they were detected at subsequent examinations, except for fQRS at ECG baseline (Appendix A). CMRI revealed an extensive subepicardial DE of inferior LVW in a non-ischemic pattern (Appendix A). Patient ID 10 sought medical attention due to three episodes of chest pain and palpitations with myocardial enzyme release, initially diagnosed as a presumed viral myocarditis. CMRI performed after the first episode showed an extensive, nearly circumferential, LGE of LVW (Appendix A) with spotty intramiocardial distribution in the inferior segment of IVS and in the mid-lateral and apical wall. This latter finding was no longer reported in the following exams. Of note, CMRI performed in occasion of these episodes failed in identifying any signs of myocardial edema. Additionally, in patients ID 15 and 18, the first clinical presentation was chest pain with myocardial enzyme release, and ECG abnormalities with normal coronary arteries. Finally, patient ID 5, unlike other patients who acceded to medical attention primarily for their myocarditis-like history, was firstly evaluated at the age of 31 years due to frequent polymorphic PVCs, absent at the peak of exercise. Both echocardiography and CMRI findings confirmed a diagnosis of DCM with left ventricular non compaction (LVNC). Particularly, CMRI revealed a moderate LVE with depressed biventricular systolic function and both intramural and subepicardial DE, respectively, of IVS and inferior LVW, midapical segments. 

Myocarditis has been previously proposed as a possible clinical presentation of ACM [31,32] and more specifically *DSP*-related cardiomyopathies [22,23,24]. Myocarditis diagnosis is often difficult due to variable clinical presentation and etiology. In most cases, due to the lack of histological data, the diagnosis is frequently based on clinical presentation, CMRI, and laboratory data, since endomyocardial biopsy is not routinary performed [33]. Therefore, it is not uncommon that myocarditis diagnosis is presumed and not proven. This applies also to myocarditis episodes reported in *DSP*-related cardiomyopathies. Moreover, some of the cases reported in scientific literature did not meet the current clinical criteria of myocarditis diagnosis. 

In 2005, Bauce et al. first described clinical myocarditis in two siblings affected by familial ARVC due to a missense variant in the *DSP* gene, already known to be associated with the disease. In both cases, chest pain and an elevation of myocardial enzymes in the absence of any abnormalities at coronary angiography were detected and reported as the first clinical symptoms [34]. Notably, in the same year, Sen-Chowdhry et al. in a cohort study of 42 patients affected by ALVC, reported that four of them presented with chest pain with enzyme release and had previously been diagnosed as affected by viral myocarditis, suggesting the possibility of myocarditis-like episodes as clinical presentation in ALVC patients [2]. The authors suggested that inflammatory myocarditis may be part of the natural history of ACM, due to genetic causes rather than infective ones. In fact, variants in the desmosomal gene, such as the *DSP* gene, may affect intercellular adhesions and/or intermediate filament function, leading to myocyte loss and inflammatory reaction followed by fibrous or fibrofatty tissue replacement. This process may present in an episodic fashion, triggered by unknown stimuli, which activate cell loss and inflammation in myocardial tissue, with a clinical presentation that has been defined ‘hot phase’ [35,36]. More recently, Smith et al. reported myocardial inflammatory episodes in 15% of their cohort among *DSP* patients. In addition, in 90% of patients with available CMRI data, the DE of LV was observed [22]. Similarly, Wang et al., in 2021, in a large cohort study of *DSP* carriers, reported a documented myocardial injury in 22% of patients, and in 90% of them CMRI showed a LGE distribution suggestive of myocardial fibrosis. The authors observed that myocardial injury also preceded the appearance of arrhythmias and heart failure [26]. Finally, in 2020, Piriou et al. selected six families, among those with a potential inherited cardiomyopathy phenotype, with at least one individual with a documented episode of acute myocarditis, and at least one individual with a cardiomyopathy or a history of SCD. Genetic testing identified in five of them a heterozygous pathogenic variant in *DSP,* and clinical evaluation confirmed a ALVC diagnosis. Histological data available in some of the *DSP* carriers showed a coexistence of slight inflammatory infiltrates, interstitial fibrosis, and the presence of viral genomes without overt systemic viral infection. 

The authors concluded that acute myocarditis should be considered as a further criterion for ACM diagnosis, and genetic testing should be performed in patients who experience acute myocarditis and have a family history of cardiomyopathy, or sudden death [37].

Our findings confirm myocardial episodic injuries as a clinical manifestation of *DSP*-cardiomyopathy, which may be the first clinical sign of disease, as occurred in seven patients with an history of presumed myocarditis.

Remarkably, all patients in this cohort presented LV DE at CMRI, even if systolic function was preserved. Notably, a larger extent of DE was observed in those patients with myocarditis-like history and, most commonly, it involved subepicardial layers of LV. Many studies demonstrated that the larger the extent of DE, the higher is the risk of SCD [38,39]. However, some studies do not attribute additional prediction value to the extent of LGE [40]. Additionally, data about the role of DE distribution remains unclear. Specifically, septal midwall LGE have been associated with a greater risk of SCD [41] and it has even been proposed as an exclusive predictor for SCD or ICD implantation [42]. On the contrary, other authors proposed the subepicardial distribution of LGE as an independent predictor for MAE [43]. Finally, some studies have not demonstrated an association between DE location and additional SCD risk [44]. Recently, Cipriani et al. clinically characterized a population of 87 ACM patients and 153 DCM patients who underwent CMRI with quantitative tissue characterization. All ACM patients presented LGE predominantly localized in the subepicardial LVW. A greater amount of LGE was observed in those patients with a decreased LVEF [45]. The authors suggested that, due to the extensive amount of LV fibrosis identified in ACM patients, the prophylactic implantation of ICD may be considered even if LVEF is not severely depressed [45].

In our cohort, 72% of patients presented a high burden of VE with frequent or very frequent PVCs of at least two distinct morphologies, consistent with an arrhythmogenic substrate. Notably, they were variably observed during exercise, rest, or recovery phase. Three patients suffered from MAE and three presented TWI in peripheral leads, with a lower frequency compared to literature data [22,26]. This could be related to the relatively small size of our cohort. Remarkably, 61% of patients (n = 11) underwent ICD implantation, nine of them for primary prevention and two for secondary prevention. This finding appears to be substantially in line with data which emerged from other larger cohort studies, which reported a little higher percentage of ICD implantation in *DSP* carriers [46,47]. 

A total of 16 LP or P variants were identified, 75% (n = 12) were truncating variants, two were missense, one a splicing variant, and one a duplication variant (Figure 3). Thirteen variants were not previously reported in literature and of them, 10 were not cited by ClinVar [27]. The variants of ID 3 [48], ID 5 and 17 [49], ID 15 [26,50], and ID 18 [51,52] were previously reported in literature (Table 1). Since most variants were truncating ones, it was not possible to establish an accurate genotype-phenotype correlation based on our data. However, based on molecular predicted consequences, it is possible to hypothesize that variants causing a loss of function of *DSP* are more likely to be implicated in a severe form of cardiomyopathy [21]. Generally, we observed that a significant number of variants (31%, n = 5) were located in exon 23, followed by exon 24 (19%, n = 3). Of note, two variants, one missense and one duplication, respectively, were identified in exon 7 (Figure 4).

It has been previously reported in literature that not truncating variants in exon 7 may alter DSP protein function by interfering with physiological binding with plakoglobin protein, leading to the loss of desmosomal cadherin-JUP complexes [53]. Notably, two patients (ID 7 and 9) were heterozygous carriers of the same novel truncating variant G1737Dfs*16. Both of them presented a history of myocardial injury with a substantially impaired LVEF and an extensive subepicardial DE at CMRI; patient ID 9 also presented frequent polymorphic EV. Another two patients (ID 5 and 17) resulted to be carriers of the same truncating variant R1951X, previously described in literature [49]. However, in this case, the patients shared few clinical features, such as polymorphic VE, reported as frequent in patient ID 5 and occasional in patient ID 17. This finding may be due to the known clinical variability among individuals carrying the same variants reported for many inherited heart disorders, and/or to the age gap between the two patients, which does not allow to exclude the possibility of a future manifestation of further shared clinical signs or symptoms. 

According to the recent Padua Criteria [7], only four patients presented sufficient requirements for the diagnosis of ALVC and two patients of ACM with biventricular involvement. Five patients were diagnosed as affected by DCM. These data suggest that in the left ventricular dominant form of ACM, genetic testing might be determinant to achieve a right diagnosis. 

## 4. Materials and Methods

### 4.1. Subject Selection

We selected a cohort of 15 probands among those patients referred to Medical Genetics Unit of Policlinico Tor Vergata Hospital of Rome from 2018 to 2022, with a suspected or confirmed diagnosis of ACM or DCM, carriers of heterozygous variants in *DSP* gene. We checked whether or not selected *DSP* variants should be reclassified according to current ACMG standards and guidelines [29]. We excluded five patients carrying *DSP* variants classified as of uncertain significance due to the lack of data about their actual pathogenicity and the impossibility to rightfully establish a genotype-phenotype correlation. We included 10 probands heterozygous carriers of *DSP* variants classified as likely pathogenic (LP) or pathogenic (P). Genomic database ClinVar, gnomAD (v2.1.1) and dbSNP [27,28,30] were consulted as well as scientific literature and data were reported. Segregation data analysis of probands’ relatives were collected when available. Further 8 probands carrying likely pathogenic (LP) or pathogenic (P) variants were collected after clinical and genetic diagnosis in other reference centers, for a total of 18 probands included in the study. 

### 4.2. Data Collection

Demographic and clinical data of *DSP* patients were retrospectively collected, including sex, age at diagnosis, personal history of major arrhythmias, family history of SCD and/or non-ischemic cardiomyopathies, and data from cardiologic evaluation of symptoms and imaging features. Indeed, patients were referred to our centre after a full cardiologic evaluation at Policlinico Casilino Hospital or San Giovanni Calibita Hospital, Fatebenefratelli, Isola Tiberina, comprehensive at least of baseline ECG, Holter ECG monitoring, transthoracic echocardiography (TTE) and/or CMRI. Coronary artery disease and/or others cardiovascular conditions potentially causative of pathologic myocardial modifications were excluded in all patients. Six probands had received their cardiac evaluation and genetic testing at the ASST Papa Giovanni XXIII Hospital of Bergamo. Data were collected in the same way as reported above. All cardiologic screenings were performed according to standards clinical protocols.

### 4.3. Genetic Analysis

The DNA was extracted with the EZ1 DNA Blood kit (Qiagen, Hilden, Germany) following manifacture’s protocol. DNA was analyzed by Nanodrop (Thermofisher, Waltham, MA, USA) to evaluate the quality by measurement of the levels of absorbance at different wavelengths. At the same time, the samples were quantified with the Qubit Fluorometer 2.0 through the use of the Qubit High Sensitivity Assay Kit (Thermofisher, Waltham, MA, USA) for determining the concentration. 

An on-demand custom panel was designed using Ion AmpliSeq Designer software 7.48 version. We selected 32 genes from literature belong to Cardiomyopathies and Channelopathies (Appendix A). To generate the libraries were necessary 15 ng of genomic DNA (gDNA), for each samples using Ion Chef System (Thermofisher), following the manufacturer’s instructions. The resultant template of 16 samples was sequenced on the Ion Torrent S5 platform (Thermofisher) using the 530 chip. Alignment and variant calling were carried out by the Torrent Suite (Thermofisher) using as human reference genome the GRCh37/hg19 version. Data were analyzed using the IonReporter software and the pathogenicity of the identified variants was evaluated according to the ACMG standards and guidelines [46]. The variants identified as VoUS, LP or P were selected for Sanger confirmation, performed using specific oligonulceotides whose sequence is shown in Appendix A. At ASST Papa Giovanni XXIII Hospital in Bergamo, genetic testing has been performed by bioinformatic selection of the cardiac gene panel (Appendix A) from whole exome data (Agilent, Santa Clara, CA, USA, Clinical Research Exome v.2) sequenced on the Illumina NovaSeq 6000 platform.

## 5. Conclusions

Despite the small size of this cohort, clinical and genetic data suggest that *DSP*-related cardiomyopathy could be considered a distinct clinical entity characterized by a high arrhythmic burden, variable LVE, and LV scar. Myocarditis-like episodes may represent the first clinical presentation of the disease. These results are in line with recent literature data and contribute to a better delineation of this emerging specific phenotype.

We propose to consider genetic testing, according to the following red-flags: (i) young adult subjects with family history of ACM/DCM and/or SCD; (ii) myocarditis-like picture characterized by chest pain, myocardial enzyme release, 12-lead electrocardiogram abnormalities with normal coronary arteries; and (iii) the presence of LGE with subepicardial distribution. Current criteria for risk stratification based mainly on depressed systolic function may not be sufficient for patients affected by *DSP*-related cardiomyopathy. The presence of extensive myocardial fibrosis could set up an arrhythmic substrate, which predispose to the development of malignant arrhythmias. Primary ICD implantation may be considered even in the absence of impaired LVEF. Further studies on a larger cohort are needed to identify more accurately distinctive phenotypic features and rule out a likely genotype-phenotype correlation.

## Figures and Tables

**Figure 1 ijms-24-02490-f001:**
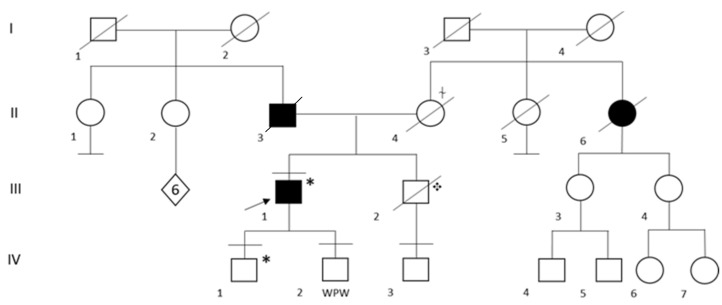
Family Pedigree of Patient ID 2. Black circle and squares indicate individuals with a clinical diagnosis of dilated cardiomyopathy. Black arrow indicates the proband. Black lines at the top indicate individuals who underwent to genetic testing. Black lines at the bottom indicate individuals without offspring. WPW: Wolf Parkinson White syndrome, ✥ Sudden Cardiac Death (SCD) at the age of 51 years. * Q833X heterozygous carrier, ⍭ Aortic dissection.

**Figure 2 ijms-24-02490-f002:**
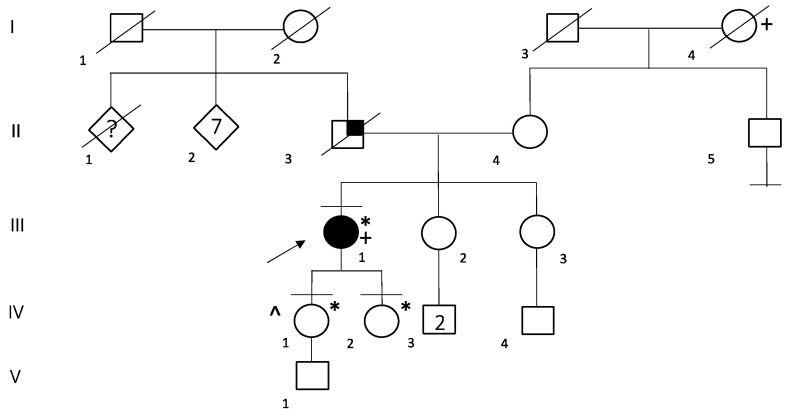
Family Pedigree of Patient ID 6. Black circle and squares indicate individuals with Arrhythmogenic cardiomyopathy. Black arrow indicates the proband. Black lines at the top indicate individuals who underwent to genetic testing. Black lines at the bottom indicate individuals without offspring. Little black square indicates lung cancer, * E1068Vfs*19 heterozygous carrier, + Breast cancer, ^ Bilateral hypoacusia, probably due to a meningitis infection on her 6th month of life.

**Figure 3 ijms-24-02490-f003:**
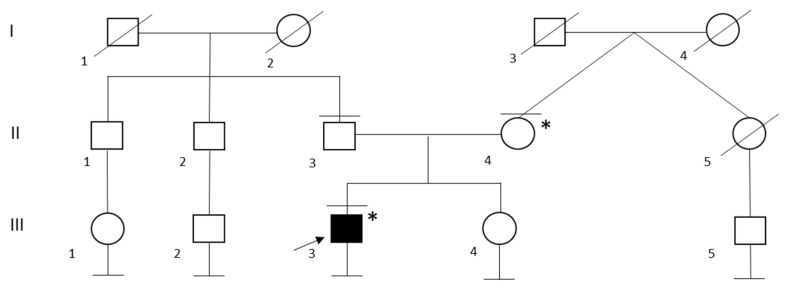
Family Pedigree of Patient ID 7. Black circle and squares indicate individuals with a clinical diagnosis of dilated cardiomyopathy. Black arrow indicates the proband. Black lines at the top indicate individuals who underwent to genetic testing. Black lines at the bottom indicates individuals without offspring. * G1737Dfs*16 heterozygous carrier.

**Figure 4 ijms-24-02490-f004:**
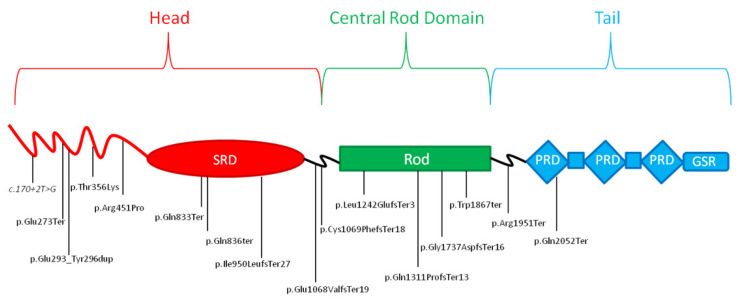
Schematic representation of the main structural domains of the desmoplakin protein and relative position of the genetic variants reported in this study. Splicing variant reported in italics. SRD:Spectrin repeat domain; PRD:plakin repeat domains; GSR:glycine/serine/arginine-rich domain.

**Table 1 ijms-24-02490-t001:** Variants identified in desmoplakin (DSP NM_004415) gene by next generation sequencing (NGS) analysis in probands.

Patients	HGVSc	HGVSp	Exon	dbSNP	gnomAD	ACMG	ClinVar	Literature
**1**	c.6154C>T	p.Gln2052Ter	24	-	-	P	-	Novel
**2**	c.2497C>T	p.Gln833Ter	18	rs1561693779	-	P	P	Novel
**3**	c.2848del	p.Ile950LeufsTer27	20	rs397516927	-	P	P	PMID: 25157032
**4**	c. 1352G>C	p.Arg451Pro	11	-	-	LP	-	Novel
**5**	c.5851C>T	p.Arg1951Ter	24	rs869025395	-	P	P/LP	PMID: 26899768
**6**	c.3203_3204del	p.Glu1068ValfsTer19	23	rs1285329067	-	P	-	Novel
**7**	c.5210del	p.Gly1737AspfsTer 16	23	rs1581819043	-	P	P	Novel
**8**	c.170+2T>G	p.(?)	-	rs1581777867	-	LP	-	Novel
**9 ***	c.5210del	p.Gly1737AspfsTer 16	23	rs1581819043	-	P	P	Novel
**10**	c.3206_3207del	p.Cys1069PhefsTer 18	23	-	-	LP	-	Novel
**11**	c.2506C>T	p.Gln836ter	18	-	-	LP	-	Novel
**12**	c.879_890dup	p.Glu293_Tyr296dup	7	-	-	LP	-	Novel
**13**	c.3724_3739del	p.Leu1242Glufs*3	23	-		LP	-	Novel
**14**	c.5601G>A	p.Trp1867ter	24	-	-	LP	-	Novel
**15**	c.1067C>A	p.Thr356Lys	9	rs780626687	1/ 251197	LP	VUS	PMID: 25227139, 34352074
**16 ***	c.3932_3936del	p.Gln1311ProfsTer13	23	-	-	LP	-	Novel
**17**	c.5851C>T	p.Arg1951Ter	24	rs869025395	-	P	P/LP	PMID: 26899768
**18**	c.816_817delinsAT	p.Gln273Ter	7	-	-	P	-	PMID: 24070718, 31402444

HGVSc: Human Genome Variation Society coding sequence name; HGVSp: Human Genome Variation Society protein sequence name; dbSNP: Single Nucleotide Polymorphism Database; gnomAD: Genomic Aggregation Database; ACMG: American College of Medical Genetics. P: Pathogenic; LP: Likely pathogenic. * Patient ID 9 and 16 carried also a heterozygous variant of uncertain significance (VoUS) in DSC2 gene (NM_024422).

**Table 2 ijms-24-02490-t002:** Clinical characteristics of probands carrying likely pathogenic or pathogenic variants in desmoplakin (DSP) gene.

Patients	Sex	Age at Diagnosis (Years)	MAEs	Syncope	Chest PAIN	Palpitations	Presumed Myocarditis	NYHA Class	ICD	FH of SCD	FH of DCM/ACM
**1**	F	40	SVT	No	Yes	Yes	No	II	Yes	Yes	DCM
**2**	M	51	No	No	No	No	No	II	Yes	Yes	DCM
**3**	F	44	No	No	No	Yes	No	I	No	No	DCM, ACM
**4**	M	32	No	No	No	No	No	I	No	No	DCM
**5**	M	31	No	Yes	No	Yes	Yes	I	Yes	Yes	No
**6**	F	57	No	No	Yes	Yes	No	II	No	No	No
**7**	M	32	No	No	Yes	No	Yes	I	No	No	No
**8**	M	31	No	No	No	No	Yes	II	Yes	No	No
**9**	F	18	No	No	Yes	Yes	Yes	I	No	No	No
**10**	M	25	No	No	Yes	Yes	Yes	I	No	Yes	No
**11**	F	46	No	No	No	Yes	No	II	Yes	Yes	Yes
**12**	F	43	No	No	No	Yes	No	I	No	Yes	DCM
**13**	M	41	No	Yes	No	Yes	No	I	Yes	Yes	ACM
**14**	F	42	No	No	No	Yes	No	II	Yes	Yes	No
**15**	F	64	No	No	Yes	Yes	Yes	II	Yes	No	DCM
**16**	F	46	SVT	No	Yes	Yes	No	I	Yes	No	No
**17**	M	20	No	No	No	Yes	No	I	Yes	No	ACM
**18**	M	68	No	No	Yes	Yes	Yes	I	Yes	Yes	No

MAEs: Major Arrhythmic Events; SVT: Sustained Ventricular Tachycardia; NYHA: New York Heart Association; ICD: Implantable cardioverter defibrillator; SCD: Sudden Cardiac Death; FH: Family history; DCM: Dilated Cardiomyopathy; ACM: Arrhythmogenic Cardiomyopathy.

**Table 3 ijms-24-02490-t003:** Cardiac imaging of probands at first clinical assessment.

Patients	LVE	RVE	LVEF	RVEF	Cardiac Wall Motion	LGE	Localization
**1**	Moderate	Moderate	45%	27%	Global hypokinesia of LV and RV	Yes	(RVW and IVS)
**2**	Severe	n.d.	22%	68%	Global hypokinesia of LV	Yes	Subendocardial with transmural extension (LVW)
**3**	n.d.	n.d.	63%	54%	Normokinesia	Yes	Subepicardial (LVW)
**4**	Mild	Mild	51%	60%	Normokinesia	Yes	Subepicardial (LVW)
**5**	Moderate	n.d.	42%	42%	Global hypokinesia of LV	Yes	Intramural (IVS), Subepicardial (LVW)
**6**	n.d.	n.d.	54%	n.a.	Normokinesia	Yes	Epicardial (LVW, RVW, IVS)
**7**	n.d.	n.d.	55%	57%	Normokinesia	Yes	Subepicardial (LVW and IVS)
**8**	Mild	n.d.	55%	51%	Hypokinesia of anterior wall and mid-apical IVS	Yes	Subepicardial (LVW and IVS)
**9**	n.d.	n.d.	63%	56%	Normokinesia	Yes	Mid-Subepicardial (LVW)
**10**	Mild	Mild	48%	43%	Global hypokinesia of LV and RV Diskynesia of RVOT	Yes	Subepicardial (LVW)
**11**	Severe	Mild	35%	51%	Global hypokinesia of LV	Yes	Transmural extension (LVW) Subepicardial (IVS, LVW)
**12**	n.d	n.d.	56%	44%	Hypokinesia of mid-apical IVS	Yes	Intramural (IVS)
**13**	n.d.	n.d.	51%	53%	Hypokinesia of inferior mid-apical, mid-lateral-wall and mid-apical IVS	Yes	Intramural (LVW), Subepicardial (LVW)
**14**	Moderate	n.d.	42%	n.d.	Hypokinesia of anterior wall and mid-apical IVS	Yes	Mid-Subepicardial (LVW)
**15**	Moderate	n.d.	35%	56%	Global hypokinesia of LV	Yes	Intramural (LVW)
**16**	Moderate	n.d.	44%	n.d.	Global hypokinesia of LV, Akinesia of anterior and infero-lateral mid-basal LVW	Yes	Transmural (LVW) Subepicardial (anterior and infero-lateral mid-basal LVW)
**17**	Mild	n.d.	52%	53%	Global hypokinesia of LV	Yes	Subepicardial (LVW, IVS and RVW)
**18**	n.d.	n.d.	60%	62%	Normokinesia	Yes	Intramural, Subepicardial (LVW)

LVE: Left Ventricular Enlargement; RVE: Right Ventricular Enlargement; LVEF: Left Ventricular Ejection Fraction; RVEF: Right Ventricular Ejection Fraction; IVS: intraventricular septum; LGE: Late Gadolinium Enhancement; LVW: Left ventricle walls; RVW: Right ventricle walls; n.d.: not detected.

**Table 4 ijms-24-02490-t004:** Electrocardiographic findings and arrhythmia burden of our cohort.

Patients	AV Conduction	IV Conduction	QRS Complex	VRA	VE	NSVT	SVT
**1**	Normal	Normal	Low voltage	TWI (Anterior and infero-lateral leads)	Frequent, Polymorphic	Yes	SVPT
**2**	Normal	LAD	PRWP	NSA	Occasional	n.d.	n.d.
**3**	Normal	RAD	Normal	Normal	Occasional	n.d.	SVPT
**4**	Normal	Normal	Normal	TWI (Inferior leads)	Very frequent, Polymorphic	Yes	n.d.
**5**	Normal	Normal	Normal	NSA	Frequent, Polymorphic	n.d.	n.d.
**6**	Normal	Normal	fQRS	NSA	Frequent, Polymorphic	n.d.	n.d.
**7**	Normal	Normal	fQRS	Normal	Occasional	n.d	n.d.
**8**	Normal	Normal	Low voltage	NSA	Frequent, Polymorphic	Yes	n.d.
**9**	Normal	Normal	Low voltage	NSA	Frequent, Polymorphic	n.d.	n.d.
**10**	Normal	LAH	Low voltage Wide QRS	Normal	Occasional, Polymorphic	n.d.	n.d.
**11**	Normal	Normal	fQRS, Q wave	Normal	Frequent, Polymorphic	n.d.	n.d.
**12**	Normal	Normal	Low voltage	NSA (infero-lateral leads)	Frequent, Polymorphic	n.d.	n.d.
**13**	Normal	EAS	Normal	NSA	Frequent, Polymorphic	Yes	n.d.
**14**	Normal	Normal	Low Voltage	NSA (infero-lateral leads)	Frequent, Polymorphic	Yes	n.d.
**15**	Normal	Normal	fQRS, Q wave	NSA	Frequent, Polymorphic	Yes	Yes
**16**	AVB I	Normal	Normal	NSA (infero-lateral leads)	Frequent Polymorphic	n.d.	Yes
**17**	Normal	Normal	Normal	Normal	Occasional Polymorphic	n.d	n.d.
**18**	Normal	Normal	fQRS	NSA	Frequent, Polymorphic	Normal	Normal

AV conduction: Atrio-ventricular conduction; IV conduction: Intraventricular conduction; VRA: Ventricular Repolarization abnormalities; VE: Ventricular Ectopia; NSVT: Non-sustained Ventricular Tachycardia; SVT: Supra-ventricular Tachycardia; LAD: Left Axial Deviation; PRWP: Poor R Wave Progression; NSA: Non-specific abnormalities; RAD: Right Axial Deviation; SVPT: Supra-ventricular Paroxysmal Tachycardia; TWI: T wave inversion; fQRS: Fragmentation of QRS complex; LAH: Left anterior hemiblock; n.d.: not detected.

**Table 5 ijms-24-02490-t005:** Data related to implantable cardioverter defibrillator (ICD) implantation.

**Patients**	**1**	**2**	**5**	**8**	**11**	**13**	**14**	**15**	**16**	**17**	**18**
**Age (years)**	64	51	36	34	47	41	43	64	46	21	68
**ICD device**	D-C	S-C	S	S	D-C	D-C	S-C	D-C	D-C	S-C	D-C
**LVEF/** **RVEF**	LVEF 48% RVEF 27%	LVEF 22% RVEF 68%	LVEF 42% RVEF 42%	LVEF 38% RVEF 51%	LVEF 35% RVEF 51%	LVEF 55% RVEF 53%	LVEF 44% RVEF 55%	LVEF 35% RVEF 54%	LVEF 44% RVEF n.d.	LVEF 50% RVEF n.d.	LVEF 60% RVEF 62%
**Time from diagnosis (years)**	24	0	5	3	1	0	0	0	0	1	0

LVEF: Left Ventricle Ejection Fraction; RVEF: Right Ventricle Ejection Fraction. D-C: Dual-Chamber; S-C: Single-Chamber; S: Subcutaneous.

## Data Availability

Some of the variants have been submitted to ClinVar (SUB12648635 clivar).

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
