# Peer review of "DSP-Related Cardiomyopathy as a Distinct Clinical Entity? Emerging Evidence from an Italian Cohort"

_ijms, 2023, doi:10.3390/ijms24032490_

Round 1

Reviewer 1 Report (Previous Reviewer 2)

The revised manuscript sounds OK. I feel it is now suitable for the journal.

This manuscript invested the association between variants in desmoplakin gene and cardiomyopathy in patients. In this study, 18 probands were included. The methods used in the study were proper and the conclusion could be supported by the experimental evidence and scientific discussion. The data were well presented and the English was good.

Author Response

we thank the reviewer for the considerations about our revised form

Reviewer 2 Report (New Reviewer)

In the manuscript 'DSP-related Cardiomyopathy as a distinct clincal entity? Emerging evidence from an Italian cohort' submitted by Di Lorenzo et al. to the International Journal of Molecular Sciences, the authors caracterize at the clinical and genetic level a cohort of different patients carrying mutations in DSP (demoplakin). The manuscript is interesting for a broad readership and describes novel aspects to arrhythmogenic cardiomyopathy. However, there are some small points which can be improved in a minor revision:

1.) All human gene names should be written in Italics in the complete manuscript.

2.) Please define all abbreviations, when they were used the first time.

3.) Line 28: How was the diagnosis of 'presumed myocarditis' done? Please definie or explain in more detail.

4.)Line 45: Please add the following citation: Francisco José Bermúdez-Jiménez et al. Circulation, 2018, PMID: 29212896 DOI: 10.1161/CIRCULATIONAHA.117.028719

In this manuscript the authors identified a novel DES mutation in a large ACM family with left-ventricular phenotype. Because DES encodes desmin as a binding partner of DSP, this study should not be ignored here.

4.) Line 63/64: Please add a review article (e.g. Gerull B and Brodehl A, 2021, Curr Heart Fail Rep 2021 Dec;18(6):378-390.) about the genetic background of ACM. 

5.) Line 67: A citation of a review article explaining the conncetion between desmosomes and intermediate filaments would be helpful (e.g. Brodehl A, 2018, Biophysical Reviews, PMID: 29926427, PMCID: PMC6082305, 

  • DOI: 10.1007/s12551-018-0429-0). 
  •  
  •  

 6.) Figure 1: What is the meaning of the slash (/) below II-2? What is the meaning of the black line below II-5?

7.) What is the meaning of the black line belwo II-5 in Figure 2?

8.) What are the meanings of the black lines below generation III?

9.) Figure 3/4. Labelling is wrong (two times Figure 3).

10.) Increase the size of this nice overview figure. Could you highlight the different domains in different colors? 

11.) Line 472/473: Please indicate the oligonucleotides used for Sanger sequencing.

12.) Acknowledgments: Please update this paragraph :-)

In total, there are several points which should be changed in the revision. However, I am optimistic that the authors can change the manuscript in a sufficient way and that this manuscript can be published afterwards. Good luck!

Author Response

please see the file attached

This manuscript is a resubmission of an earlier submission. The following is a list of the peer review reports and author responses from that submission.

Round 1

Reviewer 1 Report

In the present study, Sangiuolo and co-workers explored the clinical features of a small cohort of Italian patients carriers of heterozygous desmoplakin (DSP) pathogenic (P) or likely pathogenic (LP) variants. The authors found that the majority of DSP variants carriers suffer from ventricular arrhythmia, left ventricular enlargement and left ventricular myocardial fibrosis. Based on the presented data, the authors recommended considering DSP-related cardiomyopathy as a distinct clinical entity. The whole idea of the paper is based on this author’s recommendation which could be fine if a larger cohort was explored and a control population was implicated as well. However, with only nine patients presenting relatively variable clinical features, it seems difficult to confidently go straightaway to this recommendation.  In support, the authors should present clinical (ECG, cardiac imaging) and histological findings mentioned in the paper either in the core manuscript itself or at least as supplementary data (at least one example of each described feature). 

Although the sample collection period is mentioned (from 2018 to 2022), the chronological progress of the disease in this cohort is not clearly described. Using broad description such as “at the first clinical assessment”, “At the time of first genetic counseling” to refer to a specific time of this study is a bit confusing.

The mode of inheritance of DSP variants in this study should be specified.

The authors mentioned in the abstract that genetic data is retrospectively collected, this confuses a bit about how the patients implicated in this study are firstly selected.  

Minor comments: 

Line 62: "...which differs.." 

Line 63: "...each other for by...."

Line 76: "..some Aauthors..."

Line 78: "...requiring a its...."

Thank you

Author Response

In the present study, Sangiuolo and co-workers explored the clinical features of a small cohort of Italian patients carriers of heterozygous desmoplakin (DSP) pathogenic (P) or likely pathogenic (LP) variants. The authors found that the majority of DSP variants carriers suffer from ventricular arrhythmia, left ventricular enlargement and left ventricular myocardial fibrosis. Based on the presented data, the authors recommended considering DSP-related cardiomyopathy as a distinct clinical entity. The whole idea of the paper is based on this author’s recommendation which could be fine if a larger cohort was explored and a control population was implicated as well. However, with only nine patients presenting relatively variable clinical features, it seems difficult to confidently go straightaway to this recommendation.  In support, the authors should present clinical (ECG, cardiac imaging) and histological findings mentioned in the paper either in the core manuscript itself or at least as supplementary data (at least one example of each described feature). 

R: Thank you for your comment. We aimed to describe clinical features of this cohort based on the emerging hypothesis proposed by several studies (PMID: 32372669, 34352074, 30562115, 36043215) which support the possibility to consider DSP-related cardiomyopathy as a distinct clinical entity. We are aware of the small dimension of this cohort; nevertheless, it has to be taken into account the low prevalence of this condition and the rarity of genetic variants carried by these patients. We added additional clinical findings as supplementary data as you recommended (see Supplementary Figure 1A, 1B, 2 and 3). We rephrased the title and the conclusions in order to support this hypothesis even if studies on larger cohorts are needed to confirm these findings. We proposed to offer genetic testing in individuals presenting with the described clinical features to improve their clinical management and eventually better investigate a genotype-phenotype correlation.

Although the sample collection period is mentioned (from 2018 to 2022), the chronological progress of the disease in this cohort is not clearly described. Using broad description such as “at the first clinical assessment”, “At the time of first genetic counseling” to refer to a specific time of this study is a bit confusing.

R: Thank you for your comment. In order to clarify clinical progression of the disease we indicated the age when clinical signs or symptoms presented (see Discussion section).

The mode of inheritance of DSP variants in this study should be specified.

R: Thank you for your comment. We specified the mode of inheritance of the disease in the abstract section (at second and third line) and in the introduction section (line 80 and 81) according to your suggestions.

The authors mentioned in the abstract that genetic data is retrospectively collected, this confuses a bit about how the patients implicated in this study are firstly selected.  

R: We selected a total of 14 individuals ,among those who were referred to Medical Genetics Unit of Policlinico Tor Vergata Hospital of Rome with a suspected or confirmed diagnosis of Arrhythmogenic or Dilated Cardiomiopathy, from 2018 to 2022, and carriers of heterozygous variants in DSP gene. All of them had already received genetic counselling and they had been tested for the disease at the time of writing. After a revaluation of DSP variants according to ACMG criteria (Richards et al. 2015), none of variants had been reclassified. For the study we selected nine probands resulted heterozygous carriers of likely pathogenic or pathogenic variants. We excluded five patients because they were carriers of variants of uncertain significance (VoUS) (see as rephrased in Materials and Methods section from line 373 to 409).

Minor comments: 

Line 62: "...which differs.." 

Line 63: "...each other for by...."

Line 76: "..some Aauthors..."

Line 78: "...requiring a its...."

R: Thank you for your comments; we reviewed the text according to your suggestions (see Line 73, 74, 88 and 89).

Reviewer 2 Report

This is interesting manuscript. The results were well organized and could support the conclusion. If possible, the authors may add a diagram to show the structure of the gene and the locations of those variants. This would help the readers to understand.

Author Response

This is interesting manuscript. The results were well organized and could support the conclusion. If possible, the authors may add a diagram to show the structure of the gene and the locations of those variants. This would help the readers to understand.

R: Thank you for your comment. We appreciate your suggestion and we added a figure representing schematically DSP gene structure and the relative position of the variants identified in this cohort as you recommended (see Figure 3).

Reviewer 3 Report

Di Lorenzo et al. present a manuscript that tries to describe DSP-related Cardiomyopathy as a distinct clinical entity. Despite the introduction and discussion are overall well established, I have several concerns about their research:

Major comments:

They describe a cohort of nine patients with 8 DSP LP/P pathogenic variants. This is a very small cohort to achieve clear conclusions as they state in the manuscript. Furthermore, they selected and described these patients from a cohort that they had not described in the manuscript. Although these patients are well clinically described, there is no comparison with patients with neither arrhythmogenic nor dilated cardiomyopathy of their cohort. Thus, how could we know that these patients are really different from patients with pathogenic variants in other desmosomal genes, or patients with no LP/P variants? This is only an observational study of a very small cohort, thus the conclusions of the manuscript are not supported by data.

Minor comments:

The missense variant of the manuscript is classified as LP variant. I believe it only accomplish PM2, PM5 and PP3 ACMG criteria, unsuffcient to classified ir as LP variant.

The authors do not establish the number of cases of the cohort, only that they selected 9 cases from the period 2018-2022.

Only 2 family studies were performed. Only two additional carriers were detected, one affected and one unaffected, and not included in the study (not even the affected carrier).

Two families harbor the same variant. Since it is a novel variant, did the authors check if they are relatives?

Overall, there are insufficient data to support their conclusions.

Author Response

Di Lorenzo et al. present a manuscript that tries to describe DSP-related Cardiomyopathy as a distinct clinical entity. Despite the introduction and discussion are overall well established, I have several concerns about their research:

Major comments:

They describe a cohort of nine patients with 8 DSP LP/P pathogenic variants. This is a very small cohort to achieve clear conclusions as they state in the manuscript. Furthermore, they selected and described these patients from a cohort that they had not described in the manuscript. Although these patients are well clinically described, there is no comparison with patients with neither arrhythmogenic nor dilated cardiomyopathy of their cohort. Thus, how could we know that these patients are really different from patients with pathogenic variants in other desmosomal genes, or patients with no LP/P variants? This is only an observational study of a very small cohort, thus the conclusions of the manuscript are not supported by data.

R: Thank you for your comment. We aimed to describe clinical features of this cohort based on the emerging hypothesis proposed by several studies (PMID: 32372669, 34352074, 30562115, 36043215) which support the possibility to consider DSP-related cardiomyopathy as a distinct clinical entity. We selected a total of 14 individuals, among those who were referred to Medical Genetics Unit of Policlinico Tor Vergata Hospital of Rome with a suspected or confirmed diagnosis of Arrhythmogenic or Dilated Cardiomiopathy, from 2018 to 2022, carriers of heterozygous variants in DSP gene. All of them had already received genetic counselling and they had been tested for the disease at the time of writing. After a revaluation of DSP variants according to ACMG criteria (Richards et al. 2015), none of variants had been reclassified. We selected nine probands resulted heterozygous carriers of likely pathogenic or pathogenic variants. We excluded five patients because they were carriers of variants of uncertain significance (VoUS) (see as rephrased in Materials and Methods section from line 373 to 409). We cannot add a clinical description of those patients excluded from the study because they were carriers of VoUS and the aim of this study is clinically characterize patients affected by a DSP-related cardiomyopathy. This study is focused on DSP-related cardiomyopathy, which belong to arrhythmogenic spectrum with specific clinical features and are not described as predominant in others cardiomyopathies related to desmosomal genes. We describe this cohort in order to better correlate genotype and phenotype and to highlight the importance of a subclassification possibly influencing clinical management in the near future. We rephrased the title and the conclusions in order to support this hypothesis even if studies on larger cohorts are needed to confirm these findings.

Minor comments:

The missense variant of the manuscript is classified as LP variant. I believe it only accomplish PM2, PM5 and PP3 ACMG criteria, unsuffcient to classified ir as LP variant.

R: Thank you for your comment.

Beyond PM2, PM5 and PP3, the missense variant occurs within a mutational hot spot (Hoover et al., 2021) and thus it also accomplishes PM1 criterium and for this reason it can be evaluated as LP.

The authors do not establish the number of cases of the cohort, only that they selected 9 cases from the period 2018-2022.

R: Thank you for your comment. We selected a total of 14 individuals, among those who were referred to Medical Genetics Unit of Policlinico Tor Vergata Hospital of Rome with a suspected or confirmed diagnosis of Arrhythmogenic or Dilated Cardiomiopathy, from 2018 to 2022, and carriers of heterozygous variants in DSP gene. All of them had already received genetic counselling and they had been tested for the disease at the time of writing. After a revaluation of DSP variants according to ACMG criteria (Richards et al. 2015), none of variants had been reclassified. We selected nine probands resulted heterozygous carriers of likely pathogenic or pathogenic variants. We excluded five patients because they were carriers of variants of uncertain significance (VoUS) (see as rephrased in Materials and Methods section from line 373 to 409).

Only 2 family studies were performed. Only two additional carriers were detected, one affected and one unaffected, and not included in the study (not even the affected carrier).

R: Thank you for your comment. We proposed segregation analysis to all families of affected probands. At the time of writing only two families accepted to be tested. In the Results section, from line 217 to 218 and from 222 to 223, we better clarified segregation analysis in family of proband ID 2. The second variant carrier, his first child, was apparently unaffected and for this reason was not included in the study. His father, affected by Dilated Cardiomiopathy (DCM) was already died and not available for segregation analysis (see modified pedigree – Figure 1). In family of proband ID 7 the mother resulted carrier, but she did not present cardiac abnormalities at the echocardiography performed in the previous year.

Two families harbor the same variant. Since it is a novel variant, did the authors check if they are relatives?

R: Thank you for your comment. All patients involved in the study are unrelated as specified in line 99 and 120.

Reviewer 4 Report

In Abstract the  authors do not explain abbreviation CMRL.

# The authors do not give information what was the total number of participants who were analyzed in this study and what was the number of patients with cardiomyopathy.

# The study does not give information whether or not the probands included to the analysis were suffering from other cardiovascular diseases (for instance coronary arterial disease, hypertension)

#   The study does not give information whether the patients included to the analysis received pharmaceutical treatment and if so what was the type of the treatment, and whether the treatment could influence the ECG picture.

The group of patients is small and the ECG picture heterogenous. Without the additional information it is not possible to decide whether DSP-related cardiomyopathy is really a distinct clinical entity as the title of the study suggests.

Author Response

In Abstract the authors do not explain abbreviation CMRL.

R: Thank you for your comment. We explained the abbreviation in the abstract: Cardiac magnetic resonance imaging (CMRI).

# The authors do not give information what was the total number of participants who were analyzed in this study and what was the number of patients with cardiomyopathy.

R: Thank you for your comment. We selected a total of 14 individuals, among those who were referred to Medical Genetics Unit of Policlinico Tor Vergata Hospital of Rome with a suspected or confirmed diagnosis of Arrhythmogenic or Dilated Cardiomiopathy, from 2018 to 2022, and carriers of heterozygous variants in DSP gene. All of them had already received genetic counselling and they had been tested for the disease at the time of writing. After a revaluation of DSP variants according to ACMG criteria (Richards et al. 2015), none of variants had been reclassified. We selected nine probands resulted heterozygous carriers of likely pathogenic or pathogenic variants. We excluded five patients because they were carriers of variants of uncertain significance (VoUS) (see as rephrased in Materials and Methods section from line 364 to 399).

# The study does not give information whether or not the probands included to the analysis were suffering from other cardiovascular diseases (for instance coronary arterial disease, hypertension)

R: Thank you for your comment. All patients included in the study underwent a full cardiologic evaluation, including ECG baseline, ECG Holter 24 hours, echocardiography, ECG stress test and cardiac magnetic. Coronary artery disease using cardiac CT scan or coronary angiography was excluded in all individuals such as other cardiovascular conditions which could explain morphologic or electrographic heart abnormalities (see as rephrased in Materials and Methods section, Data collection).

# The study does not give information whether the patients included to the analysis received pharmaceutical treatment and if so what was the type of the treatment, and whether the treatment could influence the ECG picture.

R: Thank you for your comment. The patients included in the study received appropriate clinical management to their condition, therefore they were pharmacologically treated both for arrhythmias and/or impaired systolic function according to guidelines. Specifically, all of them were taking beta-blockers. However, we do not believe that the pharmaceutical treatment administered could relevantly influence ECG picture and this is why we did not specify in core text patients’ pharmacological therapy.

The group of patients is small and the ECG picture heterogenous. Without the additional information it is not possible to decide whether DSP-related cardiomyopathy is really a distinct clinical entity as the title of the study suggests.

The results coming from our group of patients confirm and support recently observations coming from larger cohort of DSP variants carrier patients (see Smith et al., Circulation. 2020) suggesting DSP-related cardiomyopathy as a distinct clinical entity compared to arrhythmogenic cardiomyopathy based on clinical correlates such as episodic myocardial injury, LV systolic dysfunction, LV fibrosis and a high incidence of ventricular arrhythmias and not based on ECG picture.

Round 2

Reviewer 1 Report

Thank you for responding to the reviewer's comments. Although the authors have supported their conclusions on findings of other studies with bigger cohorts, the absence of a control cohort and the very small size of the case cohort is still a big weakness point for this paper. I recommend authors to consider a bigger population with a control if they want to well establish a genotype-phenotype correlation or if possible to redirect the study to a family study with more families included. In its actual format, it is very difficult to adopt the suggested conclusions of this paper based on the presented data. 

Best of luck.

Reviewer 4 Report

I do not have other comments for the authors.